# Computational Insights on the Potential of Some NSAIDs for Treating COVID-19: Priority Set and Lead Optimization

**DOI:** 10.3390/molecules26123772

**Published:** 2021-06-21

**Authors:** Ayman Abo Elmaaty, Mohammed I. A. Hamed, Muhammad I. Ismail, Eslam B. Elkaeed, Hamada S. Abulkhair, Muhammad Khattab, Ahmed A. Al-Karmalawy

**Affiliations:** 1Department of Medicinal Chemistry, Faculty of Pharmacy, Port Said University, Port Said 42526, Egypt; ayman.mohamed@pharm.psu.edu.eg; 2Department of Organic and Medicinal Chemistry, Faculty of Pharmacy, Fayoum University, Fayoum 63514, Egypt; mia06@fayoum.edu.eg; 3Department of Pharmaceutical Chemistry, Faculty of Pharmacy, The British University in Egypt, Cairo-Suez Desert Road, Cairo 11837, Egypt; m.ismail.800@gmail.com; 4Department of Pharmaceutical Sciences, College of Pharmacy, AlMaarefa University, Ad Diriyah, Riyadh 13713, Saudi Arabia; ikaeed@mcst.edu.sa; 5Pharmaceutical Organic Chemistry Department, Faculty of Pharmacy (Boys), Al-Azhar University, Nasr City, Cairo 11884, Egypt; habulkhair@horus.edu.eg; 6Department of Pharmaceutical Chemistry, Faculty of Pharmacy, Horus University-Egypt, New Damietta 34518, Egypt; 7Department of Chemistry of Natural and Microbial Products, Division of Pharmaceutical and Drug Industries, National Research Centre, Cairo 12622, Egypt; mkhattab@swin.edu.au

**Keywords:** drug repurposing, SARS-CoV-2 main protease, docking, molecular dynamics, DFT calculations

## Abstract

The discovery of drugs capable of inhibiting SARS-CoV-2 is a priority for human beings due to the severity of the global health pandemic caused by COVID-19. To this end, repurposing of FDA-approved drugs such as NSAIDs against COVID-19 can provide therapeutic alternatives that could be utilized as an effective safe treatment for COVID-19. The anti-inflammatory activity of NSAIDs is also advantageous in the treatment of COVID-19, as it was found that SARS-CoV-2 is responsible for provoking inflammatory cytokine storms resulting in lung damage. In this study, 40 FDA-approved NSAIDs were evaluated through molecular docking against the main protease of SARS-CoV-2. Among the tested compounds, sulfinpyrazone **2**, indomethacin **3**, and auranofin **4** were proposed as potential antagonists of COVID-19 main protease. Molecular dynamics simulations were also carried out for the most promising members of the screened NSAID candidates (**2**, **3**, and **4**) to unravel the dynamic properties of NSAIDs at the target receptor. The conducted quantum mechanical study revealed that the hybrid functional B3PW91 provides a good description of the spatial parameters of auranofin **4**. Interestingly, a promising structure–activity relationship (SAR) was concluded from our study that could help in the future design of potential SARS-CoV-2 main protease inhibitors with expected anti-inflammatory effects as well. NSAIDs may be used by medicinal chemists as lead compounds for the development of potent SARS-CoV-2 (M^pro^) inhibitors. In addition, some NSAIDs can be selectively designated for treatment of inflammation resulting from COVID-19.

## 1. Introduction

In December 2019, a novel coronavirus disease (COVID-19) was detected initially in China. The virus outbreak took place first in Wuhan city and continued to spread worldwide. After the virus’s terrible breakthrough, the World Health Organization assessed that COVID-19 was a pandemic on 11 March 2020 [1]. By 11 April 2021, approximately 136,781,961 patients were diagnosed with COVID-19, affecting 221 countries and territories around the world with a total death toll of 2,951,955 [2]. The virus is highly contagious and lethal, especially for those with other health issues [3].

Inflammatory cytokine storm is a very common critical symptom in patients with severe COVID-19, leading to systemic inflammation and high multiple organ failure [3]. Unfortunately, there are no effective drugs—to date—that can treat COVID-19. On the other hand, the development of a new drug is not a facile process and needs a lot of time and money to ensure its safe efficacy. This is one of the challenges facing the global pharmaceutical industry. Therefore, the development of alternative tools is needed to help in overcoming the prevalence of COVID-19.

Thus, we need to pave the way for unprecedented research efforts and new eligible approaches within a short time. One of the approaches that can play an important role in fighting COVID-19 is “drug repurposing”, also called drug repositioning, re-profiling, or re-tasking. Drug repurposing is a strategy of exploring new uses for existing approved drugs that are outside the original medical indication [4]. Therefore, drug repurposing offers a lot of benefits and can be much better than developing an entirely new drug for a certain indication [5,6]. Drug repositioning is expanding in the area of rare and neglected diseases. It helps to mitigate failures in drug discovery, and in recent years, approximately one-third of the approvals have been due to drug repurposing. In drug repurposing methods, the hidden therapeutic effects of drugs are investigated using diverse approaches, including computational manners, clinical experiments, and other in vitro approaches. The implementation of data-driven drug repurposing in most cases is integrated with computational assistance [7,8].

Computational approaches are valuable and fundamental tools in drug discovery steps and the development trajectory. Several computational approaches help researchers in the discovery of new drug candidates. An example of these in silico techniques is structure-based virtual screening and molecular docking studies [9]. On the other hand, bioinformatics can be used to detect the main key amino acids at nearly the same normal conditions, and, hence, confirming the docking results and druggability will be also easier to handle. Virtual screening can then provide possible drug candidates based on the chemical nature of the drug and its target protein, saving cost and time, and integrating intellectual intervention [10].

As mentioned earlier, one of the most critical COVID-19 symptoms is the inflammatory cytokine storm. This storm results from lung cells damaged by the virus. Subsequently, a local immune response is triggered, recruiting monocytes and macrophages that release cytokines and prime adaptive T and B cell immune responses [11]. This process is capable of resolving viral infection. In some cases, however, a dysfunctional immune response occurs, which can cause severe lung and even systemic pathology. Thus, the use of anti-inflammatory drugs in the COVID-19 treatment protocol is required [11,12]. Moreover, it was also found that SARS-CoV-2 gene mutation may correlate with enhanced cytokine production, such as TNF-α and IL-6 (Figure 1), compared to that isolated from the Wuhan virus, increasing the need for anti-inflammatory drugs [13].

As they are commonly used for pain relief and inflammation cure, non-steroidal anti-inflammatory drugs (NSAIDs) can be considered as an important step in the treatment of COVID-19. Furthermore, indomethacin showed potent antiviral activity against canine coronavirus in vitro, and this activity was also observed in vivo and against human SARS-CoV [14]. Moreover, it was found that rotavirus infectivity was decreased after treatment with NSAIDs (e.g., indomethacin, ibuprofen, mefenamic acid, and ketoprofen) [15]. Besides, auranofin (Gold NSAID) showed the potential to reduce the viral reservoir of HIV (human immunodeficiency virus) in infected T-cells [16].

In vitro studies on NSAIDs revealed that some of them can partially reduce SARS-CoV-2 replication. For example, celecoxib, indomethacin, ibuprofen, ketoprofen, ketorolac, meloxicam, and piroxicam were evaluated against NRC-03-nhCoV. The aforementioned drugs exhibited promising antiviral activities with high selectivity indexes relative to cellular toxicity. Piroxicam and indomethacin exhibited the highest potency against NRC-03-nhCoV as their IC_50_ values were estimated at 8.21 and 8.51 μM, respectively [17].

Therefore, in continuation of our previous work targeting SARS-CoV-2 main protease [6,18,19,20,21,22,23,24,25] as a promising anti-SARS-CoV-2 drug target, and taking into consideration the crucial role of M^pro^ enzyme for SARS-CoV-2 replication (the main protease enzyme of SARS-CoV-2 and also known as 3C-like protease (3CL^pro^), which is responsible for the cleavage of the coronavirus polyprotein at 11 specific sites), besides the previously mentioned activity of some NSAIDs towards different viruses, our perspective in this article is targeting the M^pro^ enzyme through virtual screening of a small library of a subset of the approved NSAIDs (Figure 2) via molecular docking of the ligands on the 3D crystal structure of M^pro^ (PDB ID: 6LU7) [26]. Thus, we can investigate the best ligands that might have antiviral activity against SARS-CoV-2 or at least recommend the best NSAID members and prioritize them to be used in the treatment of the inflammatory cytokine storms accompanying some COVID-19 cases. Also, our study sheds light on some NSAID candidates as lead compounds that can be optimized in the future to be more effective against SARS-CoV-2, which was accomplished through a structure–activity relationship (SAR) study based on the obtained results.

Molecular dynamics (MD) simulations were carried out on the docked complexes of the highest-ranked docking compounds (sulfinpyrazone **2**, indomethacin **3**, and auranofin **4**) to gain a deep understanding of the affinity between the ligand and the SARS-CoV-2 main protease active site in the explicit solvent model in order to estimate the stability of the drug within the active site of the protein and consequently confirm the docking results. 

Furthermore, sulfinpyrazone and indomethacin have been extensively studied previously for their physicochemical properties [27,28,29]. On the other hand, auranofin (AF) contains thiosugar moiety in addition to gold (Au), sulfur (S), and phosphorous (P) atoms. Hence, studying the spatial and geometrical properties of AF demands a careful choice of a quantum mechanical method that can describe AF properties more accurately than other methods. Therefore, we herein conduct a comparative study on quantum mechanical methods used to calculate the physicochemical properties of AF.

## 2. Materials and Methods

Docking studies using MOE 2019 suite [30] and molecular dynamics simulation studies using the Desmond simulation package of Schrödinger LLC [31] were carried out to examine and confirm the binding affinities and modes of the 40 selected FDA-approved NSAIDs against the viral main protease of SARS-CoV-2. The co-crystallized inhibitor (N3) was used as a reference standard. 

### 2.1. Molecular Docking

#### 2.1.1. NSAIDs Preparation

The tested compounds were downloaded from (https://pubchem.ncbi.nlm.nih.gov/ last accessed on 1 April 2021) website. Their structures and the formal charges on atoms were checked by the 2D depiction, subjected to energy minimization, and the partial charges were automatically calculated. The tested compounds together with the co-crystallized ligand (N3 inhibitor) were imported into the same database and saved in the form of an MDB file for the docking calculations with the target protease. 

#### 2.1.2. Target (SARS-CoV-2 M^pro^) Preparation

Protein Data Bank was used to download the crystal structure of SARS-CoV-2 main protease (M^pro^) (PDB code 6LU7, resolution: 2.16 Å) [26]. The downloaded protein was prepared as previously described [32]. Briefly, it was protonated and hydrogen atoms were added with their standard 3D geometry. Automatic correction for any errors in the atom’s connection and the type was also applied. Site Finder was applied for selection of the same active site of the co-crystallized inhibitor using all default parameters, and dummy atoms of the pocket were then created. 

#### 2.1.3. Docking of the Tested NSAIDs to the Viral M^pro^ Binding Site

Docking of the previously prepared database composed of our tested 40 NSAIDs and the co-crystallized inhibitor N3 was performed. The general methodology was applied as described earlier where the placement methodology was specified as triangle matcher and the scoring methodology was selected as London dG. Moreover, the refinement methodology was applied as a rigid receptor and the scoring methodology was GBVI/WSA [33,34]. Briefly, the file of the prepared active site was loaded and the general docking process was initiated. The obtained poses were studied after completion and the ones having the best ligand–enzyme interactions and the most acceptable Root Mean Squared Deviation (RMSD_refine) values were selected and stored for energy calculations. In the beginning, a validation process was also performed for the target receptor by docking only the co-crystallized ligand, and low RMSD values between the docked and the crystal conformations indicated a valid performance [35,36].

### 2.2. Molecular Dynamics (MD) Simulations

The Schrödinger LLC package [31] was used to carry out the MD simulations. The simulation system was immersed in an orthorhombic box with edges at 10 Å away from the protein molecule, implementing periodic boundary conditions. The box was filled with water described by the TIP3P model [37,38]. Salt concentration was set to 0.15 M NaCl using the Desmond system builder [39]. The OPLS3 force field [40] was utilized for the protein and the ligand parameters. The MD simulations were performed for 150 ns at the NPT ensemble (constant number of particles, pressure, and temperature). The pressure was kept constant at 1 atm implementing the Martyna–Tuckerman–Klein chain coupling scheme with 2.0 ps as coupling constant. The temperature was controlled at 300 K using the Nosé–Hoover chain coupling scheme [41,42]. Coulombic interactions were calculated using a cut-off radius of 0.9 Å.

### 2.3. Quantum Mechanical Studies

All calculations were performed on the Swinburne supercomputer using GAUSSIAN 09 Revision C.01 [43]. Four hybrid functionals B3PW91 [44], CAM-B3LYP [45], PBE1PBE [46], and wB97X [47] were utilized in conjunction with the split valence and triple zeta basis set def2tzv [48,49] for the description of the gold (Au) atom, and the standard basis set 6-311G [50] for the description of all other atoms in the auranofin (AF) molecule. Energy optimization of AF was performed in three consecutive steps utilizing 6-311G, 6-311+G*, and 6-311++G** for the description of all atoms except Au. Frequency calculation for geometries obtained from each function was then performed and no imaginary frequency was detected. The electronic circular dichroism of the AF excited state was then computed using the Density-Functional Theory (td-DFT) method.

## 3. Results and Discussion

### 3.1. Docking Studies

The SARS-CoV-2 M^pro^ has a Cys–His catalytic dyad, and the inhibitor-binding site is present in a groove between domains I and II. The N3 inhibitor is fitted inside the substrate-binding pocket of SARS-CoV-2 M^pro^ showing asymmetric units containing only one polypeptide. Molecular docking simulation of N3 inhibitor **1** and the FDA-approved NSAIDs (**2**–**41**)—described in Figure 2—into M^pro^ active site was done. They were stabilized at the N3-binding site of M^pro^ by variable several electrostatic interactions (Table 1). The order of binding strength was: N3 inhibitor (**1**, docked) > sulfinpyrazone **2** > indomethacin **3** > auranofin **4** > phenylbutazone **5** > celecoxib **6** > sulfasalazine **7** > oxyphenbutazone **8** > sulindac **9** > metamizole **10** > meloxicam **11** > oxaprozin **12** > nimesulide **13** > piroxicam **14** > valdecoxib **15** > zomepirac **16** > rofecoxib **17** > etodolac **18** > tenoxicam **19** > carprofen **20** > ketoprofen **21** > tolmetin **22** > nabumetone **23** > probenecid **24** > ketorolac **25** > ibuprofen **26** > fenoprofen **27** > flurbiprofen **28** > salsalate **29** > naproxen **30** > flufenamic acid **31** > mefenamic acid **32** > diclofenac **33** > meclofenamic acid **34** > phenacetin **35** > diflunisal **36** > aurothioglucose **37** > aspirin **38** > sodium aurothiomalate **39** > paracetamol **40** > allopurinol **41**.

Many poses were obtained with better binding modes and interactions inside the receptor pocket. The poses binding to the main amino acids with the best scores and RMSD_refine values were selected. Results of scores, RMSD values, and different interactions with amino acids of the M^pro^ pocket are represented in Table 1.

The results of docking studies revealed that sulfinpyrazone **2**, indomethacin **3**, and auranofin **4** had the best binding affinities and modes against SARS-CoV-2 main protease with binding free energies of −7.12, −7.07, and −6.91 kcal/mol, respectively (Table 1). These energy values were near to that of the docked N3 inhibitor (binding energy = −9.39 kcal/mol), and concerning that the catalytic dyad of SARS-CoV-2 M^pro^ is composed of cysteine and histidine amino acids.

The detailed binding mode of N3 was as follows; the docked N3 moiety occupied the branched pocket of M^pro^, forming four hydrogen bonds with Glu166, Gln189, Ser46, and Met49 at 2.94, 3.05, 3.12, and 3.51 Å, respectively. It also formed one pi-H interaction with His41 at 4.19 Å. However, sulfinpyrazone **2** formed two H-bonds with Glu166 and His41 at 2.98 and 3.21 Å, respectively, and a pi-H interaction with Gly143 at 3.58 Å. Furthermore, indomethacin **3** showed the formation of three hydrogen bonds, one with His163 at 3.49 Å and two with Met165 at 3.89 and 4.11 Å. It also formed one H-pi interaction with His41 and another pi-H interaction with Glu166 at 3.87 and 4.30 Å, respectively. On the other hand, auranofin **4** formed five hydrogen bonds with His41, His163, Leu141, Asn142, and Gln189 at 2.90, 3.10, 3.39, 3.41, and 3.49 Å, respectively, and an H-pi interaction with His41 at 4.20 Å (Table 2).

### 3.2. Molecular Dynamics (MD) Simulations

To get a deeper insight into the stability of the best three docked compounds (**2**, **3**, and **4**), molecular dynamics simulations were carried out for 150 ns simulation time. The co-crystallized peptide ligand M^pro^ complex was also subjected to MD simulation to be accounted for as a reference, resulting in a total of four dynamics runs.

Analyses of protein RMSD (Root Mean Square Deviation) and RMSF (Root Mean Square Fluctuation) are depicted in Figure 3. Protein structure stability throughout the simulation time is measured by protein RMSD. RMSD in the four dynamics run shows stability throughout the simulation time as compared to the reference N3 complex run.

RMSF is a measure of stability per protein residues and protein local conformational changes during the simulation. Binding site residues showed minimal local conformational changes (<2 Å) when compared to the reference structure, which indicates the conformational stability of the binding site during the simulation. Both N- and C-termini showed higher RMSF, which conforms with their high flexibility due to their flexible loop structures.

RMSD analysis of each ligand (RMSD_lig) during the simulation time is depicted in Figure 3. RMSD_lig indicates the stability of the docked pose inside the protein binding site. N3 co-crystallized ligand showed the lowest RMSD, which reflects its strongest binding to the binding site, due to its high anchorage sites to the binding site amino acids. Sulfinpyrazone showed the highest stability among the three simulated drugs. This was reflected by its low RMSD_lig and low fluctuations during the simulation time, which indicate the stability of its binding pose. Indomethacin came in third in stability ranking indicated by its slightly higher RMSD_lig than sulfinpyrazone. Additionally, the high fluctuation of RMSD_lig during the simulation was another factor affecting its weaker binding than sulfinpyrazone. Finally, auranofin showed the highest deviation from its initial predicted binding pose, as indicated by its high RMSD_lig during the simulation time. RMSD_lig of auranofin reached 80 Å, which indicates that it completely abandoned the protein binding site, also indicated by its high RMSD_lig fluctuation.

Figure 4 shows snapshots of the simulation at 0, 75, and 150 ns. For sulfinpyrazone, it showed stable binding inside the protein binding site, which is indicated by its binding poses at the start, middle, and end of the simulation. Concerning indomethacin, it abandoned the binding site in the middle of the simulation but retrieved a binding pose inside the binding site at the end of the simulation. On the contrary, auranofin abandoned its initial binding pose at the middle of the simulation, which explains its high RMSD_lig, and showed a binding near the binding site at the end of the simulation.

Figure 5 shows the number of hydrogen bonds formed between each ligand and the protein during the simulation time. N3 showed the highest number of hydrogen bonds with the protein due to its higher number of hydrogen bond donor/acceptor anchorage sites with the protein. Sulfinpyrazone showed the highest average number of hydrogen bonds among the three NSAIDs with the protein during the simulation time. Indomethacin and auranofin showed a lower number of hydrogen bonds compared to sulfinpyrazone. Figure 5 also shows the complete absence of hydrogen bonds formed by auranofin during most of the simulation time.

The binding interactions histogram was calculated for each protein–ligand complex during the simulation and is depicted in Figure 6. In the case of sulfinpyrazone, the amino acids (Arg188, Gln189, Thr190, Ala191, and Gln192) had the greatest contribution to the hydrogen bonding interactions with sulfinpyrazone (40–80%), and His164 contributed mainly with hydrophobic interactions (>80%). Leu27, Ser46, Met49, Glu166, Leu167, and Ala173 also contributed with <30% to the hydrophobic interactions. Figure 7 shows the timeline heat map for the total number of contacts between each ligand and the protein. The main binding residues with sulfinpyrazone were Met165, Gln189, Thr190, Ala191, and Gln192, which maintained contacts with sulfinpyrazone throughout >85% of the simulation time.

Indomethacin formed weaker interactions than sulfinpyrazone as it was less stable inside the binding site through the simulation. This could also be observed from the timeline protein–ligand contacts, which are shown to be less than that of sulfinpyrazone (Figure 7). The main binding interactions were water bridged hydrogen bonds with Pro132, Cys145, Met165, Leu167, Pro168, Arg188, Gln189, Ala191, Ala193, Ala194, Gly195, and Thr196. Hydrophobic interactions were maintained with indomethacin through His164, Glu166, Leu167, Gly170, Val17, and Gln192. Hydrogen bonding interactions occurred the least in indomethacin–protein contacts, which were maintained by Pro168, Asp187, Arg188, Gln189, Thr190, and Thr196.

The protein–ligand contacts histogram of auranofin showed it to have the least contacts contribution percentage (<20%) throughout the simulation time. Also, the timeline heat map shows auranofin to have much fewer contacts with the protein. The major contact type was water bridged hydrogen bonds, which can be explained by its abandonment of the binding site and interaction with the protein surface by hydrogen bonds through water molecules.

The N3 molecule formed a high number of contacts with binding site amino acid residues, with an interaction fraction comparable to that of sulfinpyrazone (80%), as observed from the protein–ligand contacts histogram (Figure 6) and heat map (Figure 7). The main interacting amino acid residues were Thr24, Thr25, Thr26, His41, Ser46, Gly143, Cys145, His164, Glu166, and Gln189, with Thr26, His41, Gly143, Cys145, His164, and Glu166 contributing to most of the interactions (>60%). The main binding interactions were found to be hydrogen and bridged hydrogen bonds. The N3 inhibitor stability during the MD simulations and higher number of contacts with the protein binding site, due to its higher number of anchorage sites, reflected its superior docking score over the remaining three NSAIDs. However, the interactions of sulfinpyrazone with the protein binding site were comparable in number and strength to that of the N3 molecule.

The analysis of RMSD_lig and protein–ligand contacts diagrams together suggests sulfinpyrazone to have the most stable binding with the protein, followed by indomethacin, followed by auranofin.

### 3.3. Quantum Mechanical Studies

Auranofin (AF) is a gold-based compound containing several stereoisomeric centers. It is well known that not all functionals are designed to describe the electronic properties of all atoms in the periodic table. Therefore, we herein utilized four hybrid functionals B3PW91, CAM-B3LYP, PBE1PBE, and wB97X in combination with def2tzvp for the description of the gold (Au) atom and 6-311++G** for the description of all H, C, N, S, and P atoms. The calculated spatial properties such as bond length and bond angle were compared with the X-ray crystallographic data of AF reported by Hill and co-workers [51]. Values of the computed parameters and the reported experimental ones are listed in Table 3.

Since the gold (Au) atom is thought to be the main contributor to AF binding affinity, hence its biological activity, we mainly focused on the bond length and bond angle where the Au atom was involved. As can be seen in Table 3, the four functionals gave values for the bond length between the gold atom and phosphorous atom (Au-P) and between the gold atom and sulfur atom (Au-S) consistent with the experimental values (deviation by only 0.03 Å), where B3PW91 and PBE1PBE performed better (deviation by only 0.02 Å) than wB97X and CAM-B3LYP (deviation by only 0.03 Å). Regarding the bond angle between Au, P, and S, it was noted that B3PW91 and wB97X functionals performed better than other functionals.

Other parameters were calculated for AF and listed in Table 3 as well. There were no significant discrepancies noted for the calculated parameters except for the energy gap between the lowest unoccupied molecular orbital (LUMO) and the highest occupied molecular orbital (HOMO). Calculation of HOMO and LUMO energy is of great importance as it helps in assessing the chemical reactivity of a drug at its binding site on a protein [1,2]. HOMO energy is a measure for the electron-donating strength of a molecule during the complex formation, while LUMO energy signifies the capacity of the electron-withdrawing of a molecule. The difference in HOMO and LUMO energy, known as HOMO–LUMO gap energy, is a measure for computing the molecular reactivity and stability of the compounds (electronic excitation energy) [3]. In other words, HOMO–LUMO plays a significant role in stabilizing the interactions between drug and target protein. Hence, the orbital energy of both HOMO and LUMO and the HOMO–LUMO energy gap were calculated to estimate the chemical reactivity of the selected compounds using DFT.

B3PW91 computed the HOMO–LUMO energy gap at 4.9385 eV while it was calculated by wB97X at 9.2141 eV. The discrepancy between the two values was estimated at 4.2756 eV. We therefore anticipated seeing this discrepancy reflected by the computed electronic distribution of the outermost molecular orbitals as well.

Surprisingly, the electronic density over HOMO, LUMO, and other molecular orbitals did not show a significant difference based on the functional used, except for L+2. The diagrammatic representations of the electronic density of the outermost molecular orbitals are depicted in Figure 8. It was also noticed that the molecular electrostatic potential (MEP) map of electron density distribution at the molecular level did not show a significant discrepancy, as can be seen in Figure 8.

Since AF structure has many chiral centers, the electronic circular dichroism spectrum (ECD) was also calculated to validate the performance of hybrid functionals. Each functional gave a distinct ECD spectrum where the position and intensity of peaks differed significantly. The ECD spectrum of AF is deposited in Figure 9.

### 3.4. Structure–Activity Relationship Studies

NSAIDs can be classified according to their chemical structures [52] into: **I.** Salicylic acid derivatives: (Sulfasalazine **7**, Salsalate **29**, Diflunisal **36**, and Aspirin **38**).**II.** *p*-Amino phenol derivatives: (Phenacetin **35** and Paracetamol **40**).**III.** Pyrazolidine dione derivatives: (Sulfinpyrazone **2**, Phenylbutazone **5**, and Oxyphenbutazone **8**).**IV.** Anthranilic acid derivatives: (Flufenamic acid **31**, Mefenamic acid **32**, and Meclofenamic acid **34**).**V.** Aryl alkanoic acid derivatives:
Indole acetic acid: (Indomethacin **3**).Indene acetic acid: (Sulindac **9**).Pyrrole acetic acid: (Zomepirac **16** and Tolmetin **22**).Phenyl acetic (propionic) acid: (Oxaprozin **12**, Etodolac **18**, Carprofen **20**, Ketoprofen **21**, Ketorolac **25**, Ibuprofen **26**, Fenoprofen **27**, Flurbiprofen **28**, Naproxen **30**, and Diclofenac **33**).
**VI.** Oxicams: (Meloxicam **11**, Piroxicam **14**, and Tenoxicam **19**).**VII.** Selective COX-2 inhibitors: (Celecoxib **6**, Valdecoxib **15,** and Rofecoxib **17**).**VIII.** Gold compounds: (Auranofin **4**, Aurothioglucose **37**, and Aurothiomalate sodium **39**).**IX.** Miscellaneous: (Metamizole **10**, Nimesulide **13**, Nabumetone **23**, Probenecid **24**, and Allopurinol **41**).

So, based on their stabilities and binding scores to the SARS-CoV-2 main protease, we could identify the structure–activity relationships of the tested NSAIDs which, interestingly, showed the following results (Figure 10):(a)Concerning salicylic acid derivatives, the best activity was attained by maintaining a salicylic acid scaffold without –OH or –COOH substitution, yet it was preferable to substitute a phenyl ring at the *para* position to –OH of the salicylic scaffold to ensure the best activity (compound **7**).(b)In addition, for *p*-Amino phenol derivatives, better activity was achieved when phenolic –OH was substituted by ethyl group (compound **35**) than unsubstituted one (compound **40**).(c)For pyrazolidine dione NSAIDs, the best activity was accomplished by substitution of a pyrazolidine ring at position 4 by [2-(phenylsulfinyl)ethyl] moiety (compound **2**).(d)Moreover, studying the structure–activity relationship for anthranilic acid derivatives revealed that substitution of a phenyl ring attached to the anthranilic acid scaffold by trifluoromethyl group at position 3 attained the best activity (compound **31**).(e)Furthermore, concerning aryl acetic/propionic acid derivatives, the best activity was attained when the indole acetic acid drug was used (compound **3**).(f)For oxicams better activity was accomplished when *2H*-1,2-benzothiazine nucleus (compounds **11** and **14**) was used rather than 2,3-dihydro-4H-thieno[2,3-e] [1,2]thiazine (compound **19**).(g)On the other hand, with regards to selective cox-2 inhibitors, it worth noting that substitution of a benzenesulfonamide scaffold at position 4 with 3-trifluoromethyl pyrazole moiety (compound **6**) showed better activity than 5-methyl isoxazole moiety (compound **15**) and 5H-furan-2-one (compound **17**).(h)Additionally, for gold anti-inflammatory compounds, the best activity was attained when gold was attached to 3,4,5-triacetyloxy-6-(acetyloxymethyl) oxane-2-thiolate moiety (compound **4**).

## 4. Conclusions

This study revealed the potential of repurposing NSAIDs to bind to the active site of the SARS-CoV-2 main protease. Molecular docking studies revealed the stability and conformational flexibility of most of these drugs in the enzyme active site. Three of the screened drugs (sulfinpyrazone **2**, indomethacin **3**, and auranofin **4**) showed the strongest binding affinities and the best binding modes as well. Furthermore, molecular dynamics simulations were performed for the most promising members from the docking studies (**2**, **3**, and **4**) and confirmed our docking results as being promising SARS-CoV-2 main protease inhibitors. The quantum mechanical studies revealed that the hybrid functional B3PW91 provided a good description of the spatial parameters of AF. Moreover, NSAIDs may be used by medicinal chemists as lead compounds for the development of potent SARS-CoV-2 (M^pro^) inhibitors. As a result, we can prioritize some NSAIDs as recommended over others in the treatment of inflammation accompanying COVID-19.

## Figures and Tables

**Figure 1 molecules-26-03772-f001:**
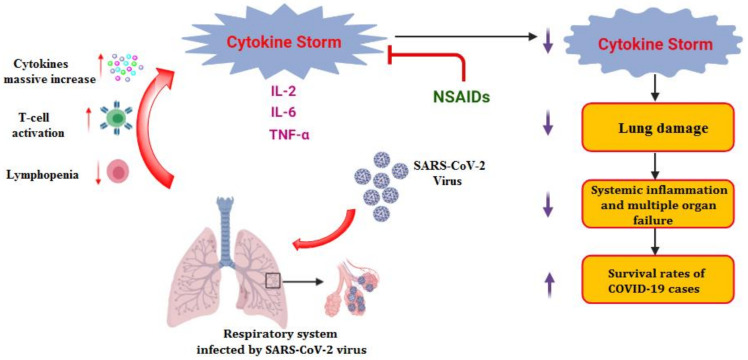
Inflammatory cytokine storm induced by SARS-CoV-2 infections and the role of anti-inflammatory drugs such as NSAIDs.

**Figure 2 molecules-26-03772-f002:**
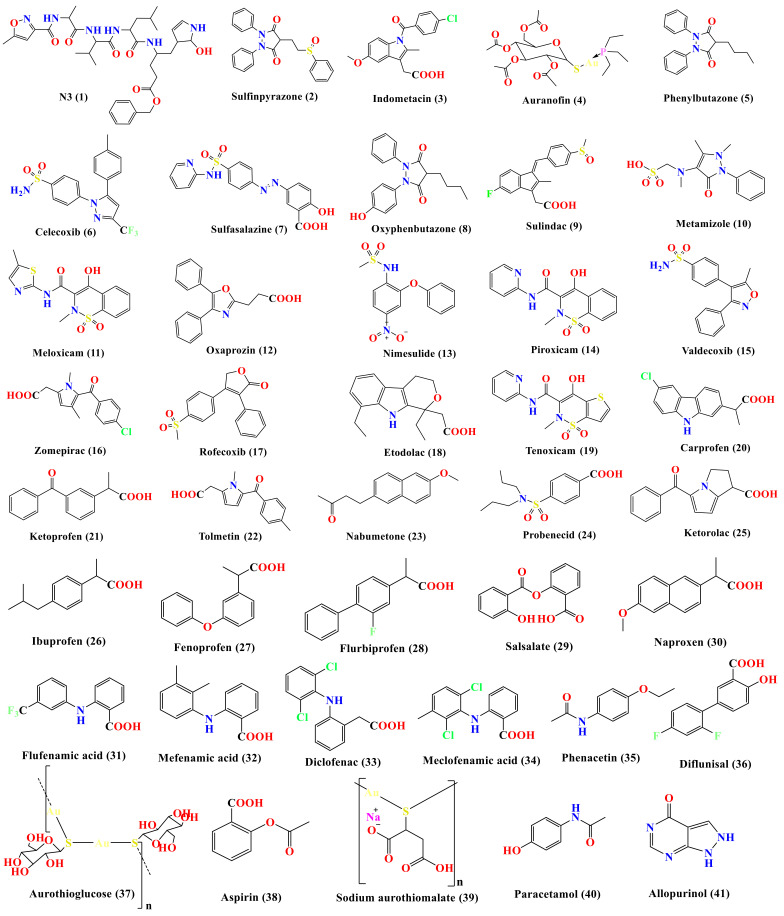
Chemical structures (in descending order of their docking scores): N3 **1**, Sulfinpyrazone **2**, Indomethacin **3**, Auranofin **4**, Phenylbutazone **5**, Celecoxib **6**, sulfasalazine **7**, Oxyphenbutazone **8**, Sulindac **9**, Metamizole **10**, Meloxicam **11**, Oxaprozin **12**, Nimesulide **13**, Piroxicam **14**, Valdecoxib **15**, Zomepirac **16**, Rofecoxib **17**, Etodolac **18**, Tenoxicam **19**, Carprofen **20**, Ketoprofen **21**, Tolmetin **22**, Nabumetone **23**, Probenecid **24**, Ketorolac **25**, Ibuprofen **26**, Fenoprofen **27**, Flurbiprofen **28**, Salsalate **29**, Naproxen **30**, Flufenamic acid **31**, Mefenamic acid **32**, Diclofenac **33**, Meclofenamic acid **34**, Phenacetin **35**, Diflunisal **36**, Aurothioglucose **37**, Aspirin **38**, Sodium aurothiomalate **39**, Paracetamol **40**, and Allopurinol **41**.

**Figure 3 molecules-26-03772-f003:**
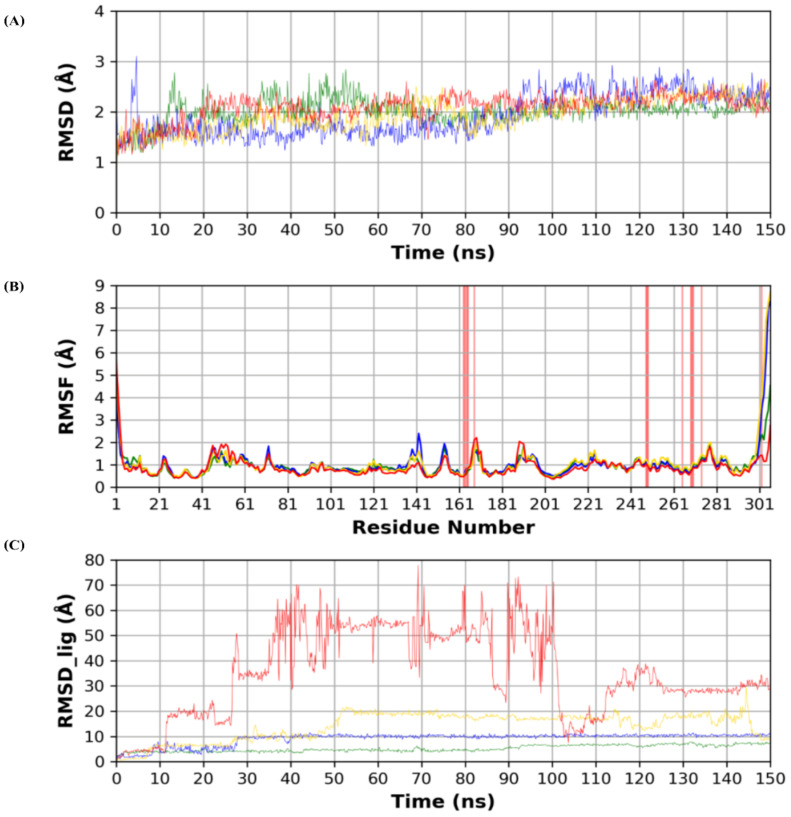
(**A**) RMSD (Root Mean Square Deviation) of the protein during the simulation time. (**B**) Per residue RMSF (Root Mean Square Fluctuation) of the protein amino acids. (**C**) RMSD of the docked poses of the four ligands inside the protein binding site. (Green: N3, Blue: Sulfinpyrazone, Yellow: Indomethacin, Red: Auranofin.)

**Figure 4 molecules-26-03772-f004:**
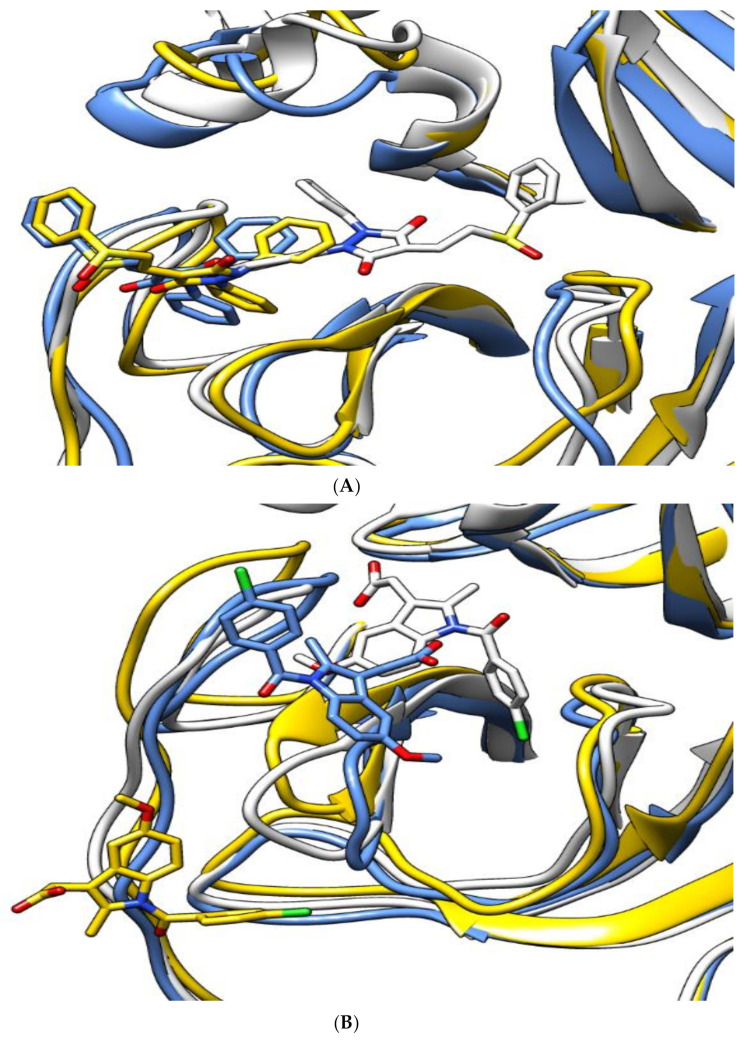
The aligned structures of protein–ligand complexes for sulfinpyrazone (**A**), indomethacin (**B**), and auranofin (**C**) during simulation. (White: 0 ns, yellow: 75 ns, blue: 150 ns.)

**Figure 5 molecules-26-03772-f005:**
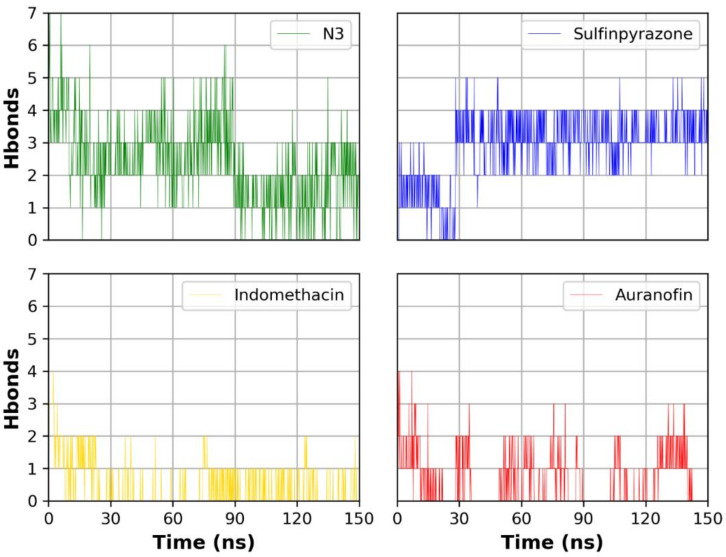
Number of hydrogen bonds formed between each ligand and the protein during the simulation.

**Figure 6 molecules-26-03772-f006:**
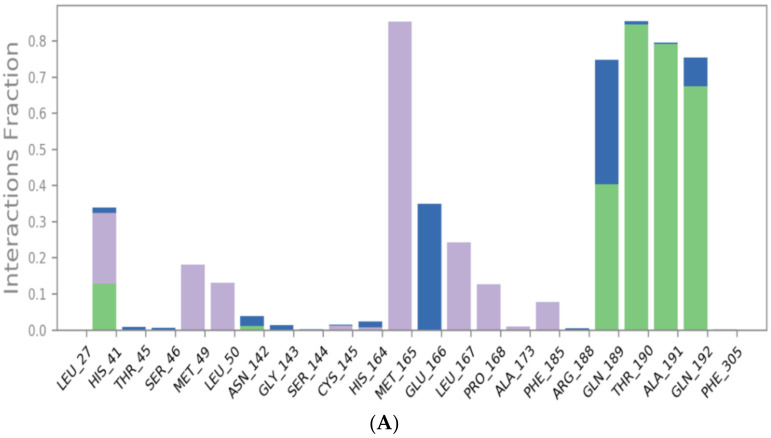
Protein–ligand contacts histograms for sulfinpyrazone (**A**), indomethacin (**B**), auranofin (**C**), and N3 (**D**).

**Figure 7 molecules-26-03772-f007:**
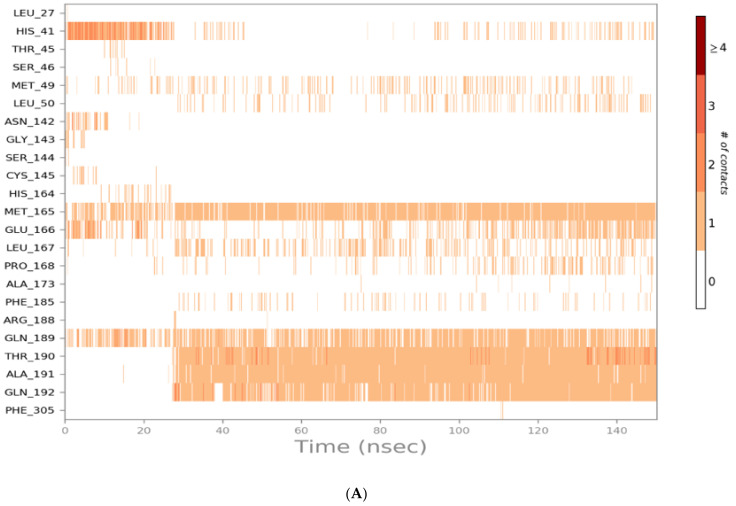
Heat map representing the number of protein–ligand contacts for sulfinpyrazone (**A**), indomethacin (**B**), auranofin (**C**), and N3 (**D**).

**Figure 8 molecules-26-03772-f008:**
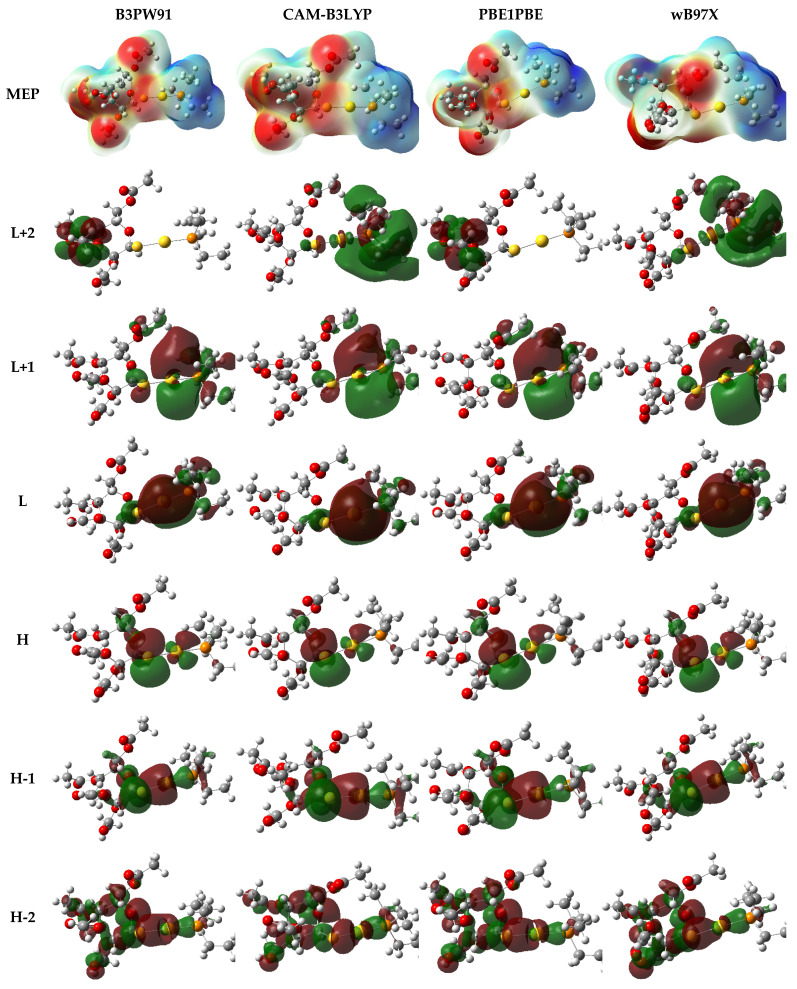
Electron density distribution of the molecular electrostatic potential (MEP) map and the outermost molecular orbitals of auranofin computed using different hybrid functionals in combination with def2tzvp and 6-311++G** basis sets.

**Figure 9 molecules-26-03772-f009:**
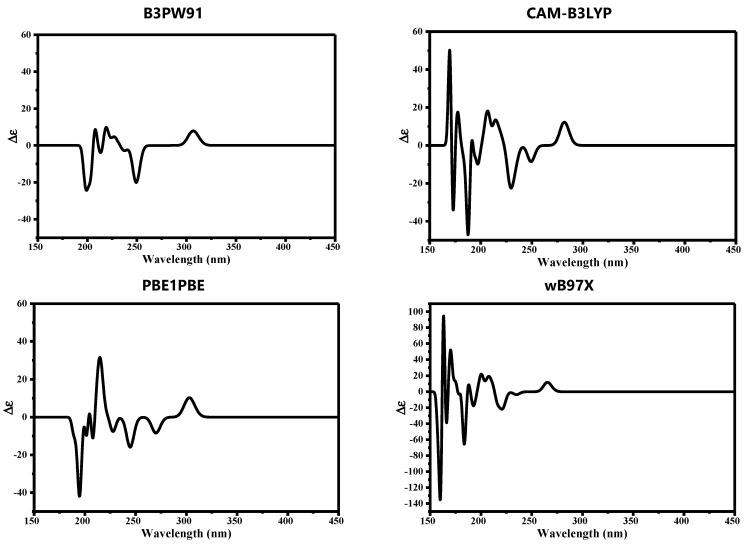
Computed electronic circular dichroism (ECD) of auranofin computed using different hybrid functionals in combination with def2tzvp and 6-311++G** basis sets.

**Figure 10 molecules-26-03772-f010:**
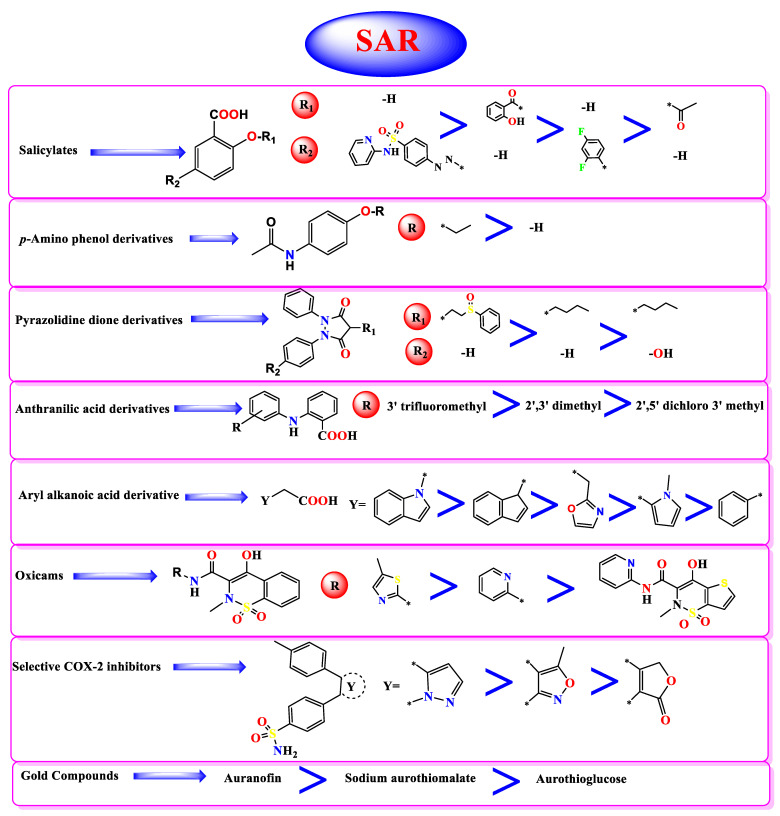
Structure-Activity Relationship (SAR) studies of the studied FDA-approved NSAIDs (**2-41**) according to their binding potentials towards the SARS-CoV-2 Mpro.*: The connection point to the main molecule.

**Table 1 molecules-26-03772-t001:** Receptor interactions and binding energies of the identified NSAIDs and N3 inhibitor (docked) into the N3 binding site of SARS-CoV-2 main protease.

No.	NSAID	S ^a^ kcal/mol	RMSD_Refine ^b^	Amino Acid Bond	Distance Å
**1**	N3	−9.39	1.78	Glu166/H-donor	2.94
				Gln189/H-acceptor	3.05
				Ser46/H-acceptor	3.12
				Met49/H-acceptor	3.51
				His41/pi-H	4.19
**2**	Sulfinpyrazone	−7.12	1.66	Glu166/H-donor	2.98
				His41/H-donor	3.21
				Gly143/H-pi	3.58
**3**	Indomethacin	−7.07	1.51	His163/H-donor	3.49
				Met165/H-acceptor	3.89
				Met165/H-acceptor	4.11
				His41/pi-H	3.87
				Glu166/H-pi	4.30
**4**	Auranofin	−6.91	0.84	His41/H-donor	2.90
				His163/H-donor	3.10
				Leu141/H-acceptor	3.39
				Asn142/H-donor	3.41
				Gln189/H-donor	3.49
				His41/pi-H	4.20
**5**	Phenylbutazone	−6.88	1.07	Glu166/H-donor	3.43
**6**	Celecoxib	−6.79	1.17	Ser144/H-donor	3.01
				His163/H-donor	3.04
				Asn142/H-acceptor	3.88
				Gln189/H-pi	4.38
**7**	Sulfasalazine	−6.76	1.77	Thr190/H-acceptor	2.82
				Glu166/H-acceptor	3.03
				Gly143/H-donor	3.14
				His41/H-donor	3.16
**8**	Oxyphenbutazone	−6.75	2.00	His164/H-acceptor	3.12
				Asn142/H-donor	3.47
				Gly143/H-donor	3.56
**9**	Sulindac	−6.67	1.25	Gly143/H-donor	2.99
				Cys145/H-donor	3.14
				Glu166/H-pi	4.49
**10**	Metamizole	−6.56	1.49	Gln189/H-acceptor	3.49
				Met165/H-acceptor	3.55
				Met165/H-acceptor	3.85
				Met165/H-pi	3.50
				His41/pi-H	4.18
				Glu166/H-pi	4.24
**11**	Meloxicam	−6.47	1.35	His163/H-donor	2.84
				His164/H-acceptor	3.15
				His164/H-acceptor	3.28
**12**	Oxaprozin	−6.43	1.20	Ser144/H-donor	2.97
				Glu166/H-pi	3.83
				Gln189/H-pi	4.03
**13**	Nimesulide	−6.35	1.35	His41/H-donor	2.86
				His163/H-donor	2.91
				Cys145/H-donor	3.46
**14**	Piroxicam	−6.30	1.32	His41/H-donor	3.30
				Cys145/H-acceptor	3.81
				Met165/H-pi	3.46
				His41/pi-H	3.54
				Glu166/H-pi	4.54
**15**	Valdecoxib	−6.30	1.12	Glu166/H-acceptor	2.89
				Met165/H-acceptor	3.40
				Gln189/H-donor	3.41
**16**	Zomepirac	−6.25	1.40	His163/H-donor	3.14
				Met165/H-pi	4.42
**17**	Rofecoxib	−6.24	1.02	Cys145/H-donor	2.99
				Met165/H-acceptor	3.48
				Asn142/H-pi	4.15
**18**	Etodolac	−6.19	0.68	Arg188/H-donor	3.28
				Glu166/H-pi	3.74
**19**	Tenoxicam	−6.18	1.47	Gly143/H-donor	2.92
				His164/H-acceptor	3.14
				Asn142/H-donor	3.18
				Gly143/H-donor	3.29
**20**	Carprofen	−6.15	0.90	His164/H-acceptor	2.95
				Gln192/H-acceptor	3.77
				Gln189/H-pi	4.53
**21**	Ketoprofen	−6.15	1.57	Glu166/H-donor	2.99
**22**	Tolmetin	−6.08	1.64	Gly143/H-donor	3.01
				His164/H-acceptor	3.08
				Cys145/H-donor	3.36
				Met49/H-acceptor	3.93
**23**	Nabumetone	−6.02	1.14	His163/H-donor	3.16
				Met165/H-pi	3.74
				Glu166/H-pi	4.16
**24**	Probenecid	−5.96	2.19	Glu166/H-donor	3.17
				Gln189/H-acceptor	3.44
**25**	Ketorolac	−5.89	1.57	Glu166/H-donor	3.05
				Glu166/H-acceptor	3.27
**26**	Ibuprofen	−5.88	0.87	Leu141/H-acceptor	2.99
				His163/H-donor	3.03
**27**	Fenoprofen	−5.84	1.14	His163/H-donor	3.01
				His163/H-donor	3.14
				Glu166/H-pi	4.04
				Met165/H-pi	4.22
**28**	Flurbiprofen	−5.74	1.03	Phe140/H-acceptor	2.91
				His163/H-donor	3.08
				Asn142/H-pi	3.82
**29**	Salsalate	−5.72	1.78	Gln189/H-acceptor	3.08
				Glu166/H-donor His41/pi-H	3.17
					3.90
**30**	Naproxen	−5.72	1.61	Gly143/H-donor	3.08
				Cys145/H-donor	3.31
**31**	Flufenamic acid	−5.70	1.26	His164/H-acceptor	2.95
				Ser144/H-donor	2.98
				Ser144/H-donor	3.09
				His164/H-acceptor	3.13
				Cys145/H-acceptor	3.21
**32**	Mefenamic acid	−5.68	2.08	Glu166/H-donor	3.06
				Gln189/H-acceptor	3.21
				Met165/H-acceptor	3.68
				Gln189/H-pi	4.05
**33**	Diclofenac	−5.54	1.66	Gln189/H-acceptor	2.89
				Glu166/H-donor	2.94
				Gly143/H-donor	3.25
				Leu141/H-acceptor	3.73
**34**	Meclofenamic acid	−5.48	1.18	Glu166/H-acceptor	2.84
				Gln192/H-donor	3.09
				Glu166/H-acceptor	3.17
**35**	Phenacetin	−5.43	1.27	Glu166/H-acceptor	3.03
				Gln189/H-donor	3.37
				His41/pi-H	4.18
**36**	Diflunisal	−5.26	1.52	Leu141/H-acceptor	2.80
				His163/H-donor	2.97
				His41/pi-H	3.83
**37**	Aurothioglucose	−4.90	1.45	His163/H-donor	3.17
				Glu166/H-acceptor	3.22
				Glu166/H-donor	3.76
				Met165/H-donor	4.08
**38**	Aspirin	−4.81	1.31	Gln189/H-acceptor	2.82
				Glu 166/H-donor	3.53
**39**	Sodium aurothiomalate	−4.67	1.42	His164/H-acceptor	2.83
				Arg188/H-donor	3.55
				Met49/H-acceptor	3.87
**40**	Paracetamol	−4.53	0.44	Glu166/H-acceptor	3.11
				Glu166/H-pi	4.25
**41**	Allopurinol	−4.33	1.13	Asp187/H-acceptor	3.24
				Gln189/H-pi	3.52

^a^ S: Score of a compound into the binding pocket of the receptor, **^b^** RMSD_Refine: Root Mean Squared-Deviation between the predicted pose (after refinement) and the crystal structure (before refinement).

**Table 2 molecules-26-03772-t002:** The 3D view of binding interactions and the 3D positioning between the tested NSAID drugs and N3-binding pocket within the SARS-CoV-2 main protease compared to the N3 (Docked).

Drug	3 D Interaction	3 D Pocket Positioning
Sulfinpyrazone **2**	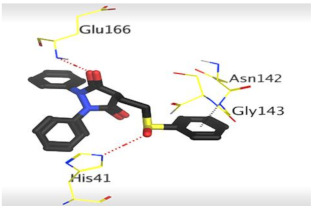	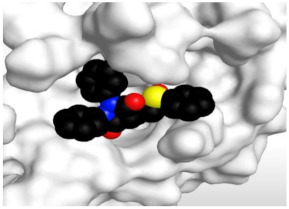
Indomethacin **3**	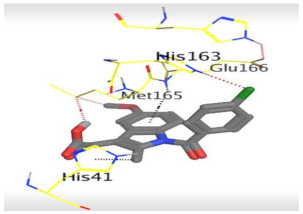	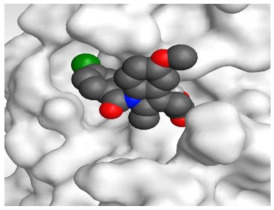
Auranofin **2**	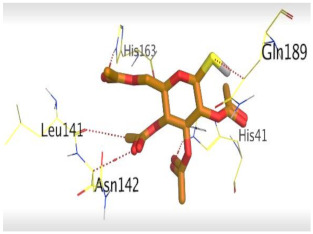	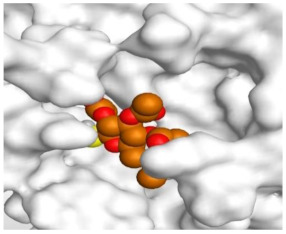
N3 **1**	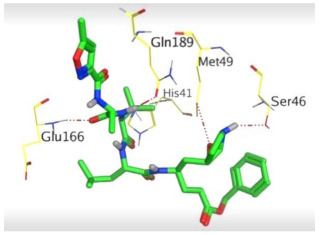	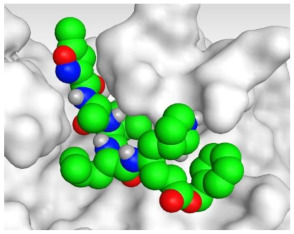

Red dashed lines refer to hydrogen bonds.

**Table 3 molecules-26-03772-t003:** Selected calculated geometric and electronic parameters of auranofin computed using different hybrid functionals in combination with def2tzvp and 6-311++G** basis sets.

	B3PW91	CAM-B3LYP	PBE1PBE	wB97X	Exp
Au-P (Å)	2.28	2.29	2.28	2.29	2.26
Au-S (Å)	2.31	2.32	2.31	2.32	2.29
∠ S-Au-P	177.4	179.2	178.9	177.4	173.6
∠ Au-S-C	102.7	101.6	101.3	102.2	105.6
E_h_ (a.u.)	−2335.103739	−2334.956583	−2333.714375	−2335.316967	-
ZPE (a.u.)	0.534394	0.540545	0.537098	0.542842	-
*E*_h_ + ZPE (a.u.)	−2334.569345	−2334.416038	−2333.177278	−2334.774126	-
Polarizability (a.u.)	348.400064	337.552355	344.065746	334.158185	-
*μ* (D)	11.5681	11.6597	11.4449	11.8367	-
HOMO-LUMO gap (eV)	4.94	7.30	5.21	9.21	-
Entropy (cal/mol-kelvin)	261.262	256.099	256.603	250.821	-

## Data Availability

The data presented in this study are available on request from the corresponding author.

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
