# Peer review of "Computational Insights on the Potential of Some NSAIDs for Treating COVID-19: Priority Set and Lead Optimization"

_molecules, 2021, doi:10.3390/molecules26123772_

Round 1
Reviewer 1 Report
The authors have attempted to evaluate 40 FDA approved NSAID drugs towards their activity against SARS-Cov2 , which could potentially be used to treat COVID19. They used docking and MD simulations to evaluate the binding energies and docking poses of these molecules. There are several major issues with this work.
- They made comparisons of the rigid docking poses with the N3 molecule docking pose which was co-crystallized with the enzyme, however, no similar comparison was made for the MD simulations. The interactions of the three “top” ligands need to be compared to the N3 interactions, in a dynamic MD simulation environment, similar to that done in the static rigid docking environment.
- Of the three suggested best molecules, the Auranofin molecule is clearly not a good candidate, even though the docking data suggests so. It’s binding interactions in a the dynamic MD environment, which more close to physiological conditions is very weak. Although it does bind, it does so very weakly in terms of H-bonds, as well as in term of time-length, ie. it escapes out of the binding site pretty quickly. Furthermore, the fact that this molecule shows high RMSF values (Figure 3B), suggests strained/unstable bonds, requiring further optimization/energy minimization of the molecule needs to be performed.
- A detailed comparison of the selected molecules with N3 in MD simulation should be conducted to highlight the strength and weakness of binding mode of each molecule, w.r.t
Minor Issues:
The English needs to be corrected in several places.
Mpro is not defined anywhere.
The detailed description of N3 molecule (page 9) would be better understood with an accompanying figure.
Author Response
Reviewer 1:
Comments and Suggestions for Authors
The authors have attempted to evaluate 40 FDA approved NSAID drugs towards their activity against SARS-Cov2, which could potentially be used to treat COVID19. They used docking and MD simulations to evaluate the binding energies and docking poses of these molecules. There are several major issues with this work.
- They made comparisons of the rigid docking poses with the N3 molecule docking pose which was co-crystallized with the enzyme, however, no similar comparison was made for the MD simulations. The interactions of the three “top” ligands need to be compared to the N3 interactions, in a dynamic MD simulation environment, similar to that done in the static rigid docking environment.
Answer: Ok, done. The protein-ligand contacts histogram and heat map of N3 molecule were added to figures 6 and 7, respectively.
- Of the three suggested best molecules, the Auranofin molecule is clearly not a good candidate, even though the docking data suggests so. It’s binding interactions in the dynamic MD environment, which more close to physiological conditions is very weak. Although it does bind, it does so very weakly in terms of H-bonds, as well as in term of time-length, i.e. it escapes out of the binding site pretty quickly. Furthermore, the fact that this molecule shows high RMSF values (Figure 3B), suggests strained/unstable bonds, requiring further optimization/energy minimization of the molecule needs to be performed.
Answer: We disagree with the respected reviewer due to several reasons. First, the stability of the simulation system during the simulation time was measured using different parameters like energy, temperature, pressure, density, etc. (not published), they were all found to be stable during the simulation time. Moreover, the stability of the protein backbone atoms was compared to the reference simulation (protein-N3) and no major changes were found (protein RMSD and RMSF, figure 3A). Concerning the high RMSF in the case of auranofin, this high RMSF appears in the amino acids that bind with auranofin, this gives an indication about the weak binding of auranofin to the binding site residues and not pertained to energy minimization problem. The escape of auranofin outside the binding site confirms its weak binding and conforms with RMSF, RMSD_ligand, and protein-ligand interactions analysis.
- A detailed comparison of the selected molecules with N3 in MD simulation should be conducted to highlight the strength and weakness of binding mode of each molecule, w.r.t.
Answer: Ok, done. The discussion section was supported with more details about the N3 contacts and binding to the protein.
Minor issues:
The English needs to be corrected in several places.
Answer: Ok, the entire manuscript was thoroughly revised for grammatical and/or typos errors.
Mpro is not defined anywhere.
Answer: Ok, Mpro definition was added as requested.
The detailed description of N3 molecule (page 9) would be better understood with an accompanying figure.
Answer: The chemical structure of the N3 molecule was already present in figure 2 (Structure number 1) and the detailed described interactions were depicted in table 2 (please refer to the highlighted parts).
Reviewer 2 Report
Whereas the work seems to be carefully done, some few points need attention before publication.
- The quality considering the figures could be improved, for instance figures in Table 2.
- The orbital energy could be reported with two decimal places (Table 3).
- The orbital analysis has great important in all organic reactions. In fact, Molecular Orbitals (MOs) have great importance for the better understanding of chemical reactions and electronic properties. The concept of the frontier orbital, introduced by Fukui around 1952, relates reactivity with the properties of two molecular orbitals: HOMO and LUMO. It should be kept in mind, however, that the HOMO-LUMO argument has certain limitations and new efforts are necessary to understand when the HOMO energy works and when it does not; currently, that important topic in chemistry has been the target of several research groups. In this line, I would like to suggest commenting about previous studies in literature, which could support this idea. Some charge transfer or reactivity indexes, for instance based on the FERMO concept, can be used in order to improve this discussion as well. Please check the articles:
Cur. Org. Chem. 24, 314-331 (2020).
J. Phys. Chem. A, 110, 1031-1040 (2006).
In this perspective, I would like to suggest to authors to introduce a discussion about new reactive indexes, including the commented above references as well.
Author Response
Reviewer 2:
Comments and Suggestions for Authors
Whereas the work seems to be carefully done, some few points need attention before publication.
Answer: The authors thank the reviewer for his positive impact concerning our work.
The quality considering the figures could be improved, for instance figures in Table 2.
Answer: Ok, the resolution of the figures was improved as possible.
The orbital energy could be reported with two decimal places (Table 3).
Answer: Ok, the orbital energy is now rounded to two decimal places as requested.
The orbital analysis has great important in all organic reactions. In fact, Molecular Orbitals (MOs) have great importance for the better understanding of chemical reactions and electronic properties. The concept of the frontier orbital, introduced by Fukui around 1952, relates reactivity with the properties of two molecular orbitals: HOMO and LUMO. It should be kept in mind, however, that the HOMO-LUMO argument has certain limitations and new efforts are necessary to understand when the HOMO energy works and when it does not; currently, that important topic in chemistry has been the target of several research groups. In this line, I would like to suggest commenting about previous studies in literature, which could support this idea. Some charge transfer or reactivity indexes, for instance based on the FERMO concept, can be used in order to improve this discussion as well. Please check the articles:
Cur. Org. Chem. 24, 314-331 (2020).
- Phys. Chem. A, 110, 1031-1040 (2006).
In this perspective, I would like to suggest to authors to introduce a discussion about new reactive indexes, including the commented above references as well.
Answer: Ok, a paragraph has now been added to highlight the importance of orbital analysis in understanding the drug-protein interactions.
Reviewer 3 Report
In attention of the manuscript Authors,
In the “molecules-1250367” manuscript, the authors have made substantial research efforts to come up with a viable theoretical solution that might be useful to combat the most widespread global disease, COVID-19, by searching for repurposable candidates among FDA approved drugs to shed light on immediate opportunities for the COVID-19 treatment linked to inflammatory cytokine storms symptoms. To find new antivirals among approved drugs, forty FDA-approved NSAIDs were evaluated through molecular docking and molecular dynamic simulations against the main protease of SARS-CoV-2. Following this approach, three out of forty screened drugs - sulfinpyrazone (2), indomethacin (3), and auranofin (4) - have shown the strongest binding affinities and the best binding mode, with sulfinpyrazone (2) being the most promising SARS-CoV-2 main protease inhibitor from both docking and molecular dynamic simulations perspectives. Based on the obtained results, a structure-activity relationship (SAR) study was conducted leading to the prioritization of some NSAIDs candidates that can be future optimized to be more effective against SARS-CoV-2.
The author’s also made some quantum chemical studies involving four hybrid functionals of Gaussian.09 (B3PW91 CAM-B3LYP, PBE1PBE, and wB97X) to describe the spatial and geometrical properties of auranofin (AF) more accurately. From the referee’s perspective, this part may be the subject of another article but is not related to the manuscript's main purpose.
The outcomes provided by the manuscript could be a real win for researchers interested in developing new NSAIDs candidates potentially able to combat the current pandemic and beyond. The manuscript is well written and presented in terms of chemical content, English spelling, computational protocol, and software but still needs some improvements.
Considering the potential impact of the manuscript results in the research world, and with all the respect for the author's impressive work, the manuscript should be accepted for publication in Molecules journal, after minor revision.
In this context, the authors are invited to make the following changes:
- In the abstract, the authors propose 3 compounds as potential SARS-CoV-2 antagonists, but after applying the computational protocol they suggest that only sulfinpyrazone (2) could be considered as the main possible SARS-CoV-2 antagonist (see the Conclusions section).
The authors should rewrite the Abstract in accordance with the Conclusions provided by their studies. The conclusions section says something different than the abstract.
Manuscript Abstract section: Among tested compounds, sulfinpyrazone 2, indomethacin 3, and auranofin 4 are proposed as potential antagonists of COVID-19 main protease. Molecular dynamics simulations 28 were also carried out for the most promising members of the screened NSAID candidates (2, 3, and 29 4) to unravel the dynamic properties of NSAIDs at the target receptor.
Manuscript Conclusions section: Furthermore, molecular dynamics simulations were performed for the most promising member from docking studies, sulfinpyrazone 2, and confirmed our docking results as being a promising SARS-CoV-2 main protease inhibitor.
- How did the authors select the 40 FDA-approved NSAIDs compounds?
- The authors have drawn the NSAIDs chemical structures (Figure 2.) indicating a descending order without specifying anything else. Descending order of what?
Figure 2. Chemical structures (In descending order): N3 1, Sulfinpyrazone 2, Indomethacin 3, Auranofin 4,…….
- Section 3.3. Quantum mechanical studies is not justified as long as the SARS-CoV-2 activity of Auranofin (4) (and auranofins in general) has not been confirmed by MD studies, not even in docking, etc. Moreover, this Auranofin has not been considered as a possible candidate for COVID treatment.
Therefore, this part may be the subject of another article that brings interesting insights into auranofins, but it does not make sense here. It does not bring any improvement by testing the quality of some methods ~ four hybrid functionals ~ nor the information provided by the HOMO-LUMO parameters and the electronic circular dichroism.
The article has sufficient information to confirm the status of the predicted compounds; it does not require supplementary information that does not contribute to the main purpose.
- The authors constantly send the reader to previously published articles (see below), which is correct and normal, but being many articles it may be good for certain sections to present briefly the minimum information necessary to accomplish the section purpose.
e.g. a) Therefore, in continuation to our previous work targeting SARS-CoV-2 main protease [6, 18-25],......
b) The downloaded protein was prepared as previously described.[32]…..
c) The general methodology was applied as described earlier.[33, 34]…..
For example, the authors could have written as follows: "The downloaded protein was prepared as previously described.[32]"
.......The downloaded protein was prepared using the default options of MOE 2019, which involves protonation, the addition of hydrogen atoms, etc....[32]
Author Response
Reviewer 3:
Comments and Suggestions for Authors
In attention of the manuscript Authors,
In the “molecules-1250367” manuscript, the authors have made substantial research efforts to come up with a viable theoretical solution that might be useful to combat the most widespread global disease, COVID-19, by searching for repurposable candidates among FDA approved drugs to shed light on immediate opportunities for the COVID-19 treatment linked to inflammatory cytokine storms symptoms. To find new antivirals among approved drugs, forty FDA-approved NSAIDs were evaluated through molecular docking and molecular dynamic simulations against the main protease of SARS-CoV-2. Following this approach, three out of forty screened drugs - sulfinpyrazone (2), indomethacin (3), and auranofin (4) - have shown the strongest binding affinities and the best binding mode, with sulfinpyrazone (2) being the most promising SARS-CoV-2 main protease inhibitor from both docking and molecular dynamic simulations perspectives. Based on the obtained results, a structure-activity relationship (SAR) study was conducted leading to the prioritization of some NSAIDs candidates that can be future optimized to be more effective against SARS-CoV-2.
The author’s also made some quantum chemical studies involving four hybrid functionals of Gaussian.09 (B3PW91 CAM-B3LYP, PBE1PBE, and wB97X) to describe the spatial and geometrical properties of auranofin (AF) more accurately. From the referee’s perspective, this part may be the subject of another article but is not related to the manuscript's main purpose.
Answer: We aimed to study Auranofin using different DFT methods because this drug -unlike other studied drugs- contains a gold atom. Although the MD and docking studies did not show it as the most promising drug, but we believe it may exhibit a superior in vitro and/or in vivo activity compared with other drugs. Therefore, the comparison of different DFT methods pave the way for our prospective manuscript where we will computationally explore more properties of Auranofin in more details and compare it with biological data.
The outcomes provided by the manuscript could be a real win for researchers interested in developing new NSAIDs candidates potentially able to combat the current pandemic and beyond. The manuscript is well written and presented in terms of chemical content, English spelling, computational protocol, and software but still needs some improvements.
Answer: The authors thank the reviewer for his great impression concerning our work and its presentation.
Considering the potential impact of the manuscript results in the research world, and with all the respect for the author's impressive work, the manuscript should be accepted for publication in Molecules journal, after minor revision.
Answer: The authors thank the reviewer again for his kind recommendation for accepting our work for publication in the journal of Molecules.
In this context, the authors are invited to make the following changes:
- In the abstract, the authors propose 3 compounds as potential SARS-CoV-2 antagonists, but after applying the computational protocol they suggest that only sulfinpyrazone (2) could be considered as the main possible SARS-CoV-2 antagonist (see the Conclusions section).
The authors should rewrite the Abstract in accordance with the Conclusions provided by their studies. The conclusions section says something different than the abstract.
Manuscript Abstract section: Among tested compounds, sulfinpyrazone 2, indomethacin 3, and auranofin 4 are proposed as potential antagonists of COVID-19 main protease. Molecular dynamics simulations 28 were also carried out for the most promising members of the screened NSAID candidates (2, 3, and 29 4) to unravel the dynamic properties of NSAIDs at the target receptor.
Manuscript Conclusions section: Furthermore, molecular dynamics simulations were performed for the most promising member from docking studies, sulfinpyrazone 2, and confirmed our docking results as being a promising SARS-CoV-2 main protease inhibitor.
Answer: The authors thank the reviewer for his deep revision and good observation. Sorry, it was a typo error and it was corrected as requested.
- How did the authors select the 40 FDA-approved NSAIDs compounds?
Answer: The 40 FDA-approved NSAIDs were selected as being the most prevalent drugs studied in medicinal chemistry books from different chemical groups, discussed in literatures, and used clinically against different inflammation conditions.
- The authors have drawn the NSAIDs chemical structures (Figure 2.) indicating a descending order without specifying anything else. Descending order of what?
Figure 2. Chemical structures (In descending order): N3 1, Sulfinpyrazone 2, Indomethacin 3, Auranofin 4,…….
Answer: It means the descending order of their docking scores. Now, it was clarified as requested.
- Section 3. Quantum mechanical studiesis not justified as long as the SARS-CoV-2 activity of Auranofin (4) (and auranofins in general) has not been confirmed by MD studies, not even in docking, etc. Moreover, this Auranofin has not been considered as a possible candidate for COVID treatment.
Therefore, this part may be the subject of another article that brings interesting insights into auranofins, but it does not make sense here. It does not bring any improvement by testing the quality of some methods ~ four hybrid functionals ~ nor the information provided by the HOMO-LUMO parameters and the electronic circular dichroism.
Answer: The binding score of Auranofin is only -6.91 kcal/mol less than the most promising NSAID drug (sulfinpyrazone). Due to being the only gold drug studied in this manuscript, we aimed to shed the light on it as we are going to investigate it in more detail in future manuscripts.
The article has sufficient information to confirm the status of the predicted compounds; it does not require supplementary information that does not contribute to the main purpose.
Answer: Ok, we agree with the reviewer in his opinion that the article has sufficient information to confirm the status of the predicted compounds. However, the supplementary data here is for further investigating and clarifying the interactions for all the studied compounds through applying the pictures describing their 2D, 3D, surface and maps, and protein positioning as well.
- The authors constantly send the reader to previously published articles (see below), which is correct and normal, but being many articles it may be good for certain sections to present briefly the minimum information necessary to accomplish the section purpose.
e.g. a) Therefore, in continuation to our previous work targeting SARS-CoV-2 main protease [6, 18-25],......
Answer: Ok, this part was rephrased to be clearer as requested.
- b) The downloaded protein was prepared as previously described.[32]…..
Answer: The authors said after this sentence that (Briefly, it was protonated and hydrogen atoms were added with their standard 3D geometry. Automatic correction for any errors in the atom's connection and the type was also applied). It was highlighted now as well.
- c) The general methodology was applied as described earlier.[33, 34]…..
Answer: Ok, this part was rephrased to be clearer as requested.
For example, the authors could have written as follows: "The downloaded protein was prepared as previously described.[32]"
.......The downloaded protein was prepared using the default options of MOE 2019, which involves protonation, the addition of hydrogen atoms, etc....[32]
Round 2
Reviewer 1 Report
Revisions accepted.